

# Time domain reflectometry (TDR) for dielectric characterization of olive mill wastewater (OMW) contaminated soils

Alessandro Comegna[a*], Antonio Coppola[a], Giovanna Dragonetti[b]

[a]School of Agricultural Forestry Food and Environmental Sciences (SAFE), University of Basilicata, Potenza, Italy.
[b]Mediterranean Agronomic Institute, Land and Water Division, IAMB, Bari, 70010, Italy.

*Correspondence to*: Alessandro Comegna (alessandro.comegna@unibas.it)

**Abstract**

Olive mill wastewater (OMW) is a compound originating from oil mills during oil extraction processes. In the Mediterranean area, more than 30 million $m^3$ of OMW are produced each year, which represents 95-97% of world production. Such volumes of untreated OMW are usually directly disposed of into drainage systems, water bodies (such as streams, lagoons and ponds), or else are sprinkled on soils, causing potentially severe environmental problems to soils and groundwater. There is thus a serious waste management problem related to the olive oil industry, such practices no longer being acceptable. In the case of on-land OMW disposal, characterization and identification of this contaminant in soils is a fundamental task especially with a view to maintaining the integrity and quality of agroecosystems. In recent years, soils have been extensively studied to detect contaminants by using various geophysical methods. Among such techniques, time domain reflectometry (TDR) has shown, in different contexts, evident sensitivity and resolution capability for characterizing contaminated soil sites. In order to further exploit the potential of the TDR technique, in the present study we conducted a series of laboratory-controlled tests to explore how OMW influences the dielectric response of contaminated soils. The research led to the development of an empirical dielectric model to estimate the presence of OMW in variably saturated-contaminated soils with different textures and pedological features.

## 1. Introduction

The olive oil industry is one of the chief agricultural sectors in the Mediterranean basin. Every year about 2 million tons of olive oil are produced (Piotrowska et al., 2011), and this production is regularly increasing (Caputo et al., 2013; Sahraoui et al., 2015).

The extraction process of olive oil generates olive mill wastewater (OMW) which is a mixture of vegetation water initially present in the drupes and the water used during the different stages of oil extraction (Colarieti et al., 2006; Sahraoui et al., 2015). The volumes of OMW produced depend on the extraction method (i.e. traditional pressing, or two-phase/three-phase centrifugation systems) and may vary between 40 and 100 liters per 100 kg of processed olives (Kavvadias et al., 2014).




OMW is a waste product with a high pollution load. It is generally characterized by a low pH, high salinity and organic
content, high chemical and biological oxygen demand, a high concentration of suspended solids, and abundant presence
of mineral elements especially nitrogen, phosphorus, potassium, calcium and magnesium (Mekki et al., 2006).
Furthermore, considerable concentrations of phenolic compounds may be detectable in this wastewater, such
concentrations usually varying between 1.0 and 10 g/l (Capasso et al., 1992; Piotrowska et al., 2011).
Due to its complex composition, OMW cannot be directly added to domestic wastewater treatment plants (Caputo et al.,
2013), and there is a lack of practical and sustainable alternative solutions to OMW disposal. This aspect represents a
potential environmental problem for olive oil-producing countries (Kavvadias et al., 2014). One solution adopted for
OMW discharge which has been legally regulated in several countries (e.g. Italy under *Legislative Decree 152/2006*) is
its use for soil fertilization. However, the benefits conferred by this practice are questionable due to its proven toxic effect
on the soil biota (Isidori et al., 2005). Furthermore, long-term OMW application may cause severe alteration of soil
chemical and physical properties.
For all the above reasons, the problem of evaluating the spatial and temporal distribution of OMW in situ represents a
research topic of great interest. It can now be dealt with, for example, by using non-invasive geophysical methods
(Huisman et al., 2003; Robinson et al., 2003).
Starting from the findings of Comegna et al. (2016), in this study we show the suitability of the TDR technique in
determining the presence of OMW in a contaminated medium. Indeed, we observed that OMW affected the dielectric
behavior of the contaminated soil. A direct dependence of the bulk electrical conductivity ($EC_b$) on OMW concentration
was experimentally documented. This dependence was investigated in depth and exploited to develop, calibrate and
validate a dielectric logarithmic model, which provides, under different levels of soil contamination, the possibility of
quantifying the presence of OMW.
**2. Dielectric permittivity and electrical conductivity determination using TDR**
TDR allows concomitant determination of soil bulk dielectric permittivity ($\varepsilon_b$) and soil bulk electrical conductivity ($EC_b$)
on the same observation volume (Dalton et al., 1984). The $\varepsilon_b$ determined by TDR requires measurement of the propagation
velocity and attenuation of an applied electromagnetic wave along a transmission line in the soil (Topp et al., 1980). At
TDR frequencies between 200 MHz to 1.5 GHz, the dielectric losses can be assumed to be negligible, and $\varepsilon_b$ along a
wave-guide line of length $L$ is a function of the propagation velocity $v$ (=$2L/t$) according to:

$$\varepsilon_b = \left(\frac{c}{v}\right)^2 = \left(\frac{ct}{2L}\right)^2 \tag{1}$$



where $c$ (=$3 \times 10^8$m s$^{-1}$) is the velocity of an electromagnetic wave in vacuum, and $t$ is the travel time, that is the time
that the TDR signal requires to travel to and from the wave-guide.
Attenuation of the TDR signal can also be used as a measure of $EC_b$. According to the *thin section approach*, originally
proposed by Giese and Tiemann (1975), $EC_b$ can be calculated as follows:

$$EC_b = \frac{\varepsilon_0 c}{L} \frac{Z_0}{Z_c} \left( \frac{2V_0}{V_f} - 1 \right) \tag{2}$$

where $\varepsilon_0$ is the dielectric permittivity of free space, $Z_0$ is the characteristic probe impedance, $Z_c$ is the TDR cable tester
output impedance, $V_0$ is the incident pulse voltage, and $V_f$ is the return pulse voltage at relatively long distances along
the waveform (Or et al., 2004).
**3. Volumetric OMW content determination in soils**
Detection of contaminants in multiphase soil systems by means of geophysical methods is problematic even if the
pollutant is homogeneously distributed within the soil matrix (Redman and De Ryck, 1994; Persson and Berndtsson,
2002; Haridy et al., 2004; Moroizumi and Sasaki, 2006; Francisca and Montoro, 2012, amongst others). The TDR
technique has the potential to reveal the presence of a contaminant in soils (see Comegna et al., 2013a; Comegna et al.,
2016, Comegna et al., 2017; Comegna et al., 2019). However, as the TDR waveform only returns *"aggregate"*
information that depends on all the distinct phases involved (Comegna et al., 2016), the challenge is to find a way to
extrapolate the dielectric weight of the pollutant from the whole dielectric response (Comegna et al., 2013b).
In the present research, we followed the same methodological approach as that of Comegna et al. (2016), which was
developed to detect and quantify the presence of organic contaminants such as non-aqueous phase liquids (NAPLs) in
variable saturated soils. We observed that the presence of an NAPL in the soil affected the dielectric response of the
medium in terms of bulk dielectric permittivity ($\varepsilon_b$ decreases as the amount of NAPL increases). Analysis of dielectric
NAPL behavior allowed us to establish a univocal relationship between the amount of NAPL in the contaminated soil
($\theta_{NAPL}$), the bulk dielectric permittivity of the multiphase medium ($\varepsilon_b$), and the final value of the reflection coefficient
($\rho_f$) which, as known, can only be determined at long TDR-travel times (Or et al. 2004). Starting from these findings, we
concentrated our efforts on OMWs, which are fluids with dielectric characteristics quite unlike those of NAPLs.
In the case of OMWs, we observed that their presence in soils scarcely alters the global dielectric response of the medium
in terms of permittivity, which for increasing amounts of OMW, varies randomly (see section 5.1 below). By contrast, at
higher propagation times (i.e. those useful for TDR-$EC_b$ calculation), a functional relationship between $\theta_{OMW}$ and $EC_b$
can be hypothesized. Such considerations allowed us to develop a logarithmic relationship between $EC_b$, calculated in the
contaminated medium, and the so-called relative volume of OMW in water ($\beta$):



$$\beta = a\, ln(EC_b) + b \qquad (3)$$

where $a$ and $b$ are coefficients which have to be experimentally determined, and the relative volume of OMW in water,
$\beta$, is defined as (Rinaldi and Francisca, 2006):

$$\beta = \frac{\theta_{OMW}}{(\theta_w + \theta_{OMW})} = \frac{\theta_{OMW}}{\theta_f} \qquad (4)$$

where $\theta_f$ and $\theta_w$ are respectively the volumetric content of the whole fluid phase and the volumetric water content. Values
of $\beta$ vary in the range between 0 for a soil-water mixture and 1 for a soil-OMW mixture.
Substituting equation 4 into equation 3, $\theta_{OMW}$ can be calculated as:

$$\theta_{OMW} = \theta_f[a\, ln(EC_b) + b] \qquad (5)$$

We observed that, for a selected soil, coefficients $a$ and $b$ depend on $\theta_f$ values (see section 5.3 below), in the sense that
for each $\theta_f$ a pair of $a$ and $b$ parameters can be estimated. Further data examination coupled with statistical analysis based
on an ANCOVA test, conducted at a significance level of $\alpha=0.05$ (for more details see Comegna et al., 2016), allowed us
to assume the coefficient $a$ of equation 5 to be constant ($a = a_c = cost$, thus independent of $\theta_f$), whereas the term $b$ can
be related to $\theta_f$ via a second-order polynomial equation:

$$b = b_1\theta_f^2 + b_2\theta_f + b_3 \qquad (6)$$

where $b_1$, $b_2$ and $b_3$ are fitting parameters of the equation.
As a result, $\theta_{OMW}$ can be finally written as follows:

$$\theta_{OMW} = \theta_f[a_c + ln(EC_b) + (b_1\theta_f^2 + b_2\theta_f + b_3)] \qquad (7)$$

Using Equation 7 $\theta_{OMW}$ may be estimated once the bulk electrical conductivity ($EC_b$) and the volumetric fluid content
($\theta_f$) of the contaminated medium are determined.
**4. Materials and methods**
**4.1 Soil and OMW properties**
The soils selected to conduct the present research were a loam *Eutric Cambisol* (IUSS Working Group WRB, 2006) and
a silt-loam *Anthrosol* (IUSS Working Group WRB, 2006, both of which are found in southern Italy. Table 1 reports the
main physical and chemical properties of the two soils, while Table 2 shows a characterisation of the OMW employed in
the laboratory experiments.
Total polyphenol content was obtained using the Folin-Ciocalteu colorimetric method (APHA, 1995). Absorbance was
measured at 760 nm with a SpectroVis Plus (Vernier Software & Technology) UV-visible spectrophotometer. Total
nitrogen (TN), total organic content (TOC) and chemical oxygen demand (COD) were determined by using the IRSA-





CNR 4060 method (IRSA-CNR, 2003), the IRSA-CNR 5040 method (IRSA-CNR, 2003) and the IRSA-CNR 5130
method (IRSA-CNR, 2003), respectively.

**4.2. Experimental equipment**

The experimental apparatus consists of a TDR unit (Tektronix 1502C cable tester) and a three-wire TDR probe (with
wave guides 14.5 cm long) connected via an RG58 coaxial cable to the tester. The TDR signals once acquired were post-
processed for $\varepsilon_b$ and $EC_b$ calculation with a homemade Matlab code. The laboratory system used during the experiments
is illustrated in Figure 1.

**4.3. Laboratory experiments**

The laboratory experiments were carried out on repacked soil samples. Simultaneous measurements of $\varepsilon_b$ and $EC_b$ have
been made on soil samples that were adequately prepared as a mix of known amounts of soil and volumetric water ($\theta_w$)
and OMW ($\theta_{OMW}$) content, following the scheme of table 3. Soil samples were oven dried at 105°C and sieved at 2 mm.
The different combinations of soil, water and OMW were mixed and then kept for 24 hours in plastic bags to ensure that
OMW and water were uniformly distributed within the soil. Since the TDR signal (hence the dielectric response of a
medium) is influenced by soil porosity $\phi$ (see, for example, Jung et al., 2013), soil samples were cautiously placed in
plastic cylindrical containers (15 cm high and 9.5 cm in diameter) until the bulk densities of 1.27 g cm$^{-3}$ (*Eutric Cambisol*)
and 1.13 g cm$^{-3}$ (*Anthrosol*) were reached. Finally, a TDR probe was inserted vertically into the samples. The same
procedure was replicated on a second set of samples for model validation. The laboratory tests were conducted at a
constant temperature of 25°C.

**4.4. Model performance evaluation**

Three statistical indices were selected and calculated for evaluating model performance (equation 7): i) mean absolute
percentage error (*MAE*), ii) model efficiency (*EF*), and iii) maximum absolute percentage error (*ME*), determined
according to the following relations (Legates and McCabe Jr, 1999; Goovaerts et al., 2005):

$$MAE(\%) = \frac{\sum_{i=1}^{N}|E_i - O_i|}{N} \cdot 100 \tag{8}$$

$$EF = 1 - \frac{\sum_{i=1}^{N}(E_i - O_i)^2}{\sum_{i=1}^{N}(O_i - \overline{O})^2} \tag{9}$$

$$ME(\%) = MAX|E_i - O_i| \cdot 100 \tag{10}$$

where $E_i$ is the prediction (model-simulated data) and $O_i$ is the true value (observed data), $\overline{O}$ is the mean of the observed
data, and $N$ is the number of observations.



**5. Results and discussion**
**5.1 Dielectric characterization of OMW-contaminated soil**
Figures 2a and b show respectively the experimental $\varepsilon_b$ vs $\theta_f$ and $EC_b$ vs $\theta_f$ relationships, obtained for selected $\beta$ values.
As can be observed in Figure 2a, in the observed $\theta_f$ domain (i.e. $0.05 \leq \theta_f \leq 0.40$), the measured dielectric permittivity of
OMW-contaminated soil samples increases overall as the volumetric fluid content increases. At the same time, for fixed
$\theta_f$ values, it may be noted that the calculated $\varepsilon_b$ values more or less overlap. This means that differences in $\beta$ (i.e.
differences in soil contamination levels) do not affect the dielectric response of the contaminated medium in terms of
permittivity. In other words, $\varepsilon_b$ is not OMW-sensitive. By contrast, on observing the graphs in figures 2b, especially in
the $\theta_f$ range 0.20-0.40, a clear correlation appears between $EC_b$ and $\theta_f$ and, for a fixed $\theta_f$, between $EC_b$ and $\beta$. Indeed,
$EC_b$ values increase with $\theta_f$ and with $\beta$.
**5.2 Model calibration and validation**
In order to confirm the approach adopted, as described in section 3 above, figures 3a and b show the experimental (colored
dots) and inferred (continuous line) $\beta$ vs $\ln(EC_b)$ relationships for different values of the volumetric fluid content ($\theta_f$). On
such data, an ANCOVA analysis performed at a significance level of 0.05 confirmed a parallelism among the $\beta$-$\ln(EC_b)$
regression lines. As a consequence, a common slope $a_c$ can be assumed for each of the tested soils. Furthermore, as
demonstrated by figures 4a and b, the intercepts $b$ of the different $\beta$-$\ln(EC_b)$ relationships can be suitably inferred from a
second order polynomial equation ($R^2$ is 1.0 for the *Eutric Cambisol* and 0.99 for the *Anthrosol*). Coefficients $a_c$, $b_1$, $b_2$
and $b_3$ resulting from model calibration are shown in table 4.
As mentioned above, model reliability was evaluated by applying the model with the calibrated coefficients to an
independent validation dataset. Figure 5 compares the computed (equation 7) and the measured volumetric OMW content.
The corresponding statistical indices are reported in table 5. Overall, both figure 5 and table 5 confirm the satisfactory
agreement of the model predictions with the experimental data: model efficiency is very close to 1 for both soils;
maximum absolute percentage error and mean absolute error are, respectively, 8.8% and 3.4% for the *Eutric Cambisol*
and 6.5% and 2.8% for the *Anthrosol*.
Considering the complexity of the modeled process, these results are appreciable and validate the scientific consistency
of the approach and its general applicability to determining volumetric OMW content in a contaminated medium by
means of TDR.
**6. Conclusions**
In the present study, we conducted a series of laboratory experiments on soil samples subjected to variable degrees of
OMW contamination. Measurements of soil bulk dielectric permittivity ($\varepsilon_b$) and soil bulk electrical conductivity ($EC_b$)



were simultaneously taken, via TDR, within each investigated sample. The experimental framework was set up in order
to accomplish, as far as possible, a full factorial plan of electromagnetic characterization of the OMW-contaminated soil
samples in the $0.05 \leq \theta_f \leq 0.40$ domain. It was shown that the presence of olive mill wastewater in the soil had a low or
null effect on $\varepsilon_b$. However, an interesting correlation between $\theta_{OMW}$ and $EC_b$ was found. On the basis of the results attained,
a dielectric model (equation 7) which allows the volumetric OMW content to be quantified was developed and
appropriately validated. The research in question can be considered an enhancement in monitoring soil affected by OMW
contamination using the time domain reflectometry technique.
In order to expand the available data set, further experiments should be conducted, for example in other pedological
contexts. Full field-scale tests should also be carried out to evaluate the performance of the proposed model in real field
conditions.
*Data availability*. The dataset used in this paper is available on request to alessandro.comegna@unibas.it.
*Competing interests*: The authors declare that they have no conflict of interest.

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





**Tables**

Table 1. Main physico-chemical properties of the two soils investigated.

| Soil | Depth (cm) | Soil Texture and Classification (USDA) | | | | Porosity (%) | $C$ (%) | $EC_w$ (dSm$^{-1}$) | pH |
|---|---|---|---|---|---|---|---|---|---|
| | | Sand (%) | Clay (%) | Silt (%) | | | | | |
| E. Cambisol | 0-20 | 41.4 | 16.4 | 42.2 | Loam | 0.52 | 0.30 | 0.13 | 8.40 |
| Anthrosol | 0-20 | 15.7 | 11.6 | 72.7 | Silt Loam | 0.57 | 1.84 | 0.17 | 8.37 |

Table 2. Main physico-chemical properties of the OMW used in the experimentation.

| Parameter | Value |
|---|---|
| pH | 3.85 |
| Electrical conductivity at 20°C (dS/m) | 10.20 |
| Dissolved oxygen: $DO$ (mg/l) | 0.23 |
| Total organic carbon: TOC (mg/l) | 6016 |
| Total N (mg/l) | 650 |
| Chemical oxygen demand: COD (mg/l) | 65000 |
| Total polyphenols (mg/l) | 1718 |

Table 3. Combinations of moisture volume ($V_w$) and OMW volume ($V_{OMW}$) for $\beta$ and $\theta_f$ values.

| $\theta_f$ | Volume of fluids (cm$^3$) | Relative volume of OMW in water: $\beta$ | | | | | $\theta_f$ | Volume of fluids (cm$^3$) | Relative volume of OMW in water: $\beta$ | | | | |
|---|---|---|---|---|---|---|---|---|---|---|---|---|---|
| | | 1 | 0.75 | 0.50 | 0.25 | 0.10 | | | 1 | 0.75 | 0.50 | 0.25 | 0.10 |
| 0.05 | $V_w$ | 0 | 13 | 27 | 40 | 48 | 0.25 | $V_w$ | 0 | 66 | 133 | 199 | 239 |
| | $V_{OMW}$ | 53 | 40 | 27 | 13 | 5 | | $V_{OMW}$ | 266 | 199 | 133 | 66 | 27 |
| 0.10 | $V_w$ | 0 | 27 | 53 | 80 | 96 | 0.30 | $V_w$ | 0 | 80 | 159 | 239 | 287 |
| | $V_{OMW}$ | 106 | 80 | 53 | 27 | 11 | | $V_{OMW}$ | 319 | 239 | 159 | 80 | 32 |
| 0.15 | $V_w$ | 0 | 40 | 80 | 120 | 144 | 0.35 | $V_w$ | 0 | 93 | 186 | 279 | 335 |
| | $V_{OMW}$ | 159 | 120 | 80 | 40 | 16 | | $V_{OMW}$ | 372 | 279 | 186 | 93 | 37 |
| 0.20 | $V_w$ | 0 | 53 | 106 | 159 | 191 | 0.40 | $V_w$ | 0 | 106 | 213 | 319 | 383 |
| | $V_{OMW}$ | 213 | 159 | 106 | 53 | 21 | | $V_{OMW}$ | 425 | 319 | 213 | 106 | 43 |

Table 4. Estimated $a_c$, $b_1$, $b_2$ and $b_3$ coefficients of $\beta$ vs $\ln(EC_b)$ relationships at different $\theta_f$ values.

| Soil | $a_c$ | $b_1$ | $b_2$ | $b_3$ |
|---|---|---|---|---|
| Eutric Cambisol | 1.185 | -16.103 | -1.367 | 2.989 |
| Anthrosol | 1.569 | -22.646 | 4.7463 | 1.927 |

Table 5. Range of model applicability and: i) mean absolute error ($MAE$), ii) maximum absolute percentage error ($ME$), iii) model efficiency ($EF$), referring to measured and predicted (equation 7) volumetric OMW content ($\theta_{OMW}$).

| Soil | Range of model applicability | MAE (%) | ME (%) | EF |
|---|---|---|---|---|
| Eutric Cambisol | $0.20 \leq \theta_f \leq 0.40$ | 3.4 | 8.80 | 0.95 |
| Anthrosol | $0.20 \leq \theta_f \leq 0.40$ | 2.8 | 6.53 | 0.96 |





**Figures**

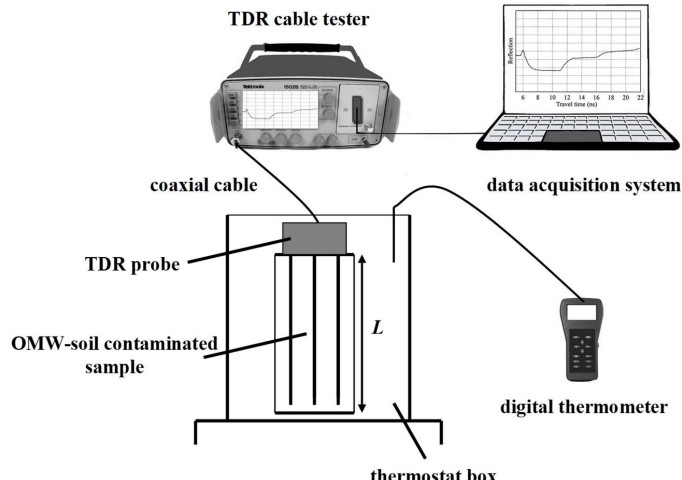

**Figure 1.** Experimental setup used in laboratory experiments (from Comegna et al., 2016).



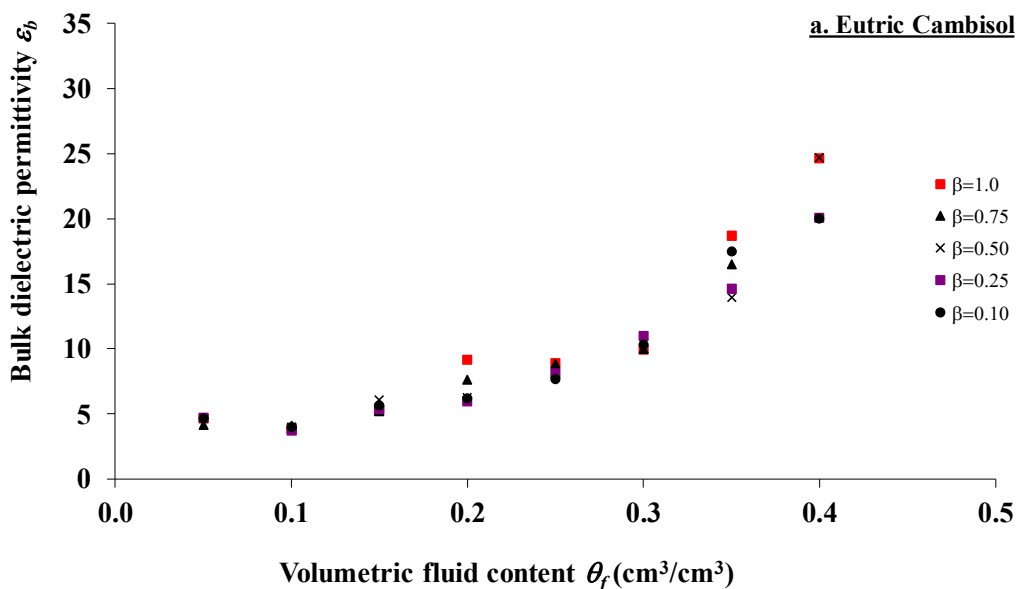

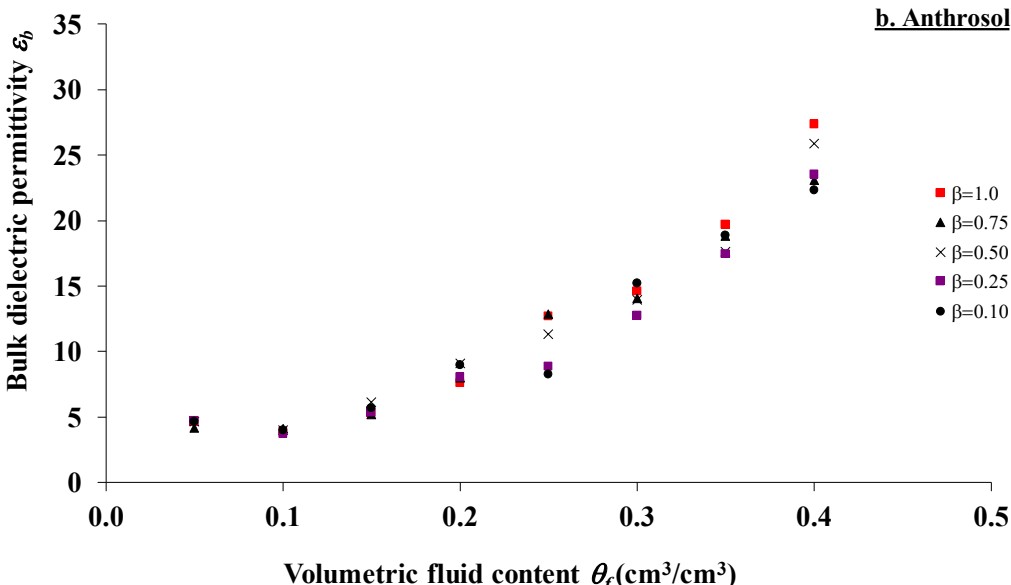



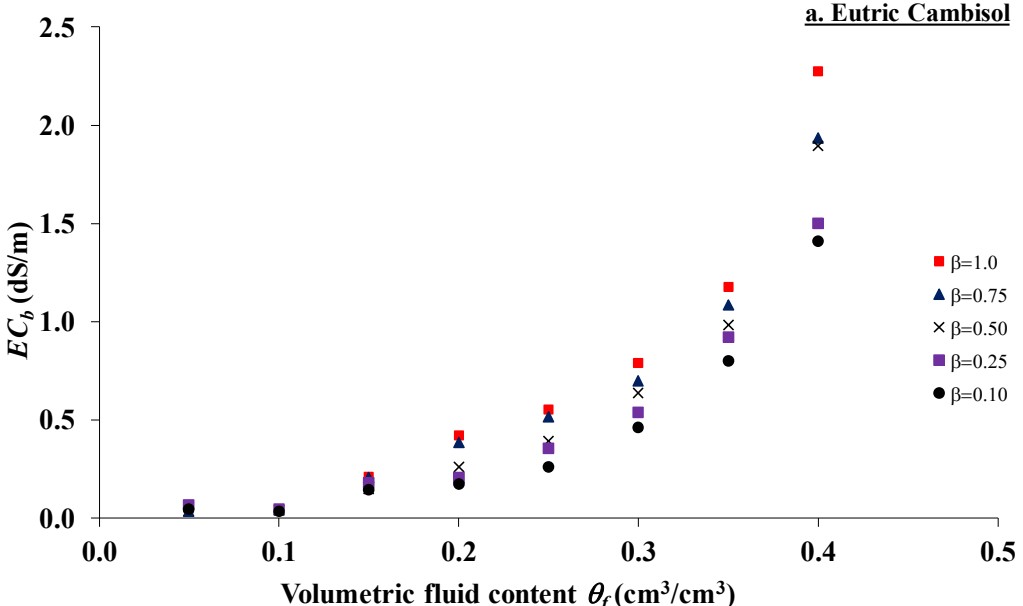

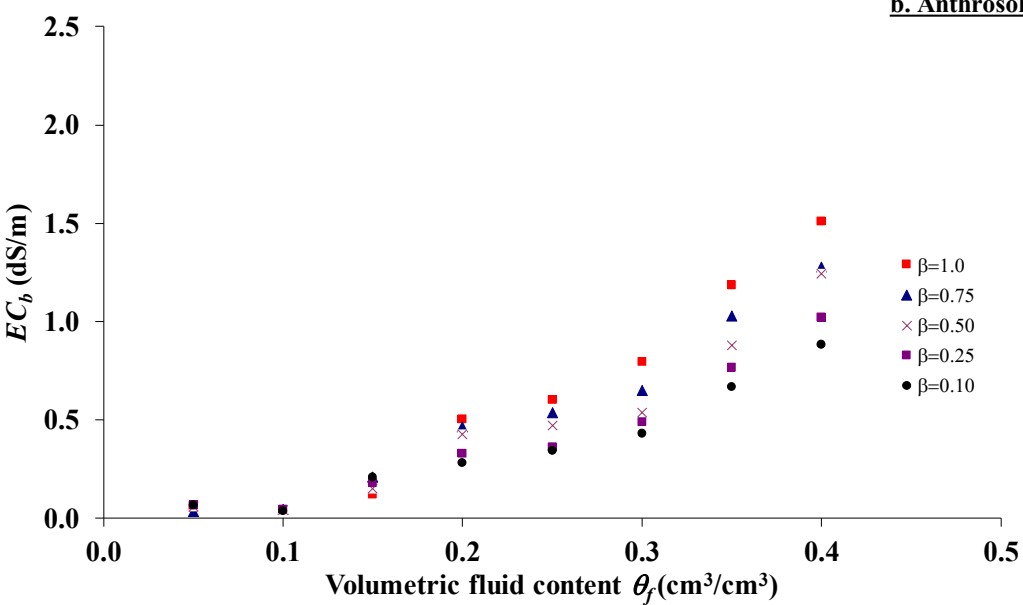

**Figure 2.** Effect of volumetric fluid content ($\theta_f$) on: a) bulk dielectric permittivity ($\varepsilon_b$), and b) bulk electrical conductivity

($EC_b$), of soil-water-OMW-air mixtures, for different $\beta$ values.



**Figure 3** Experimental relationship between bulk electrical conductivity $EC_b$ and the relative volume of OMW in water, for constant $\theta_f$ values: a) *Eutric Cambisol* and b) *Anthrosol*.

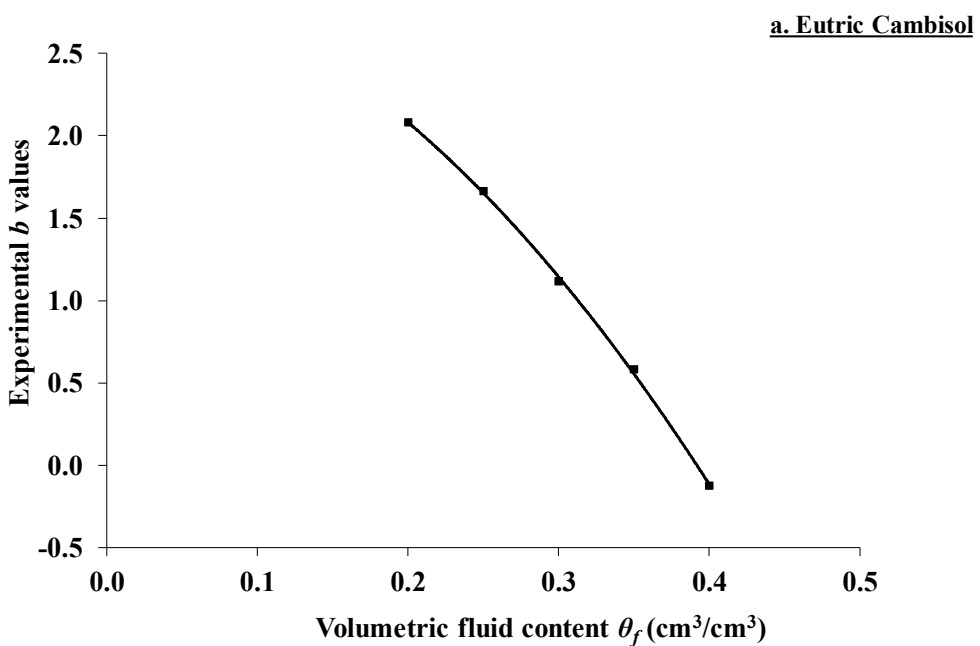

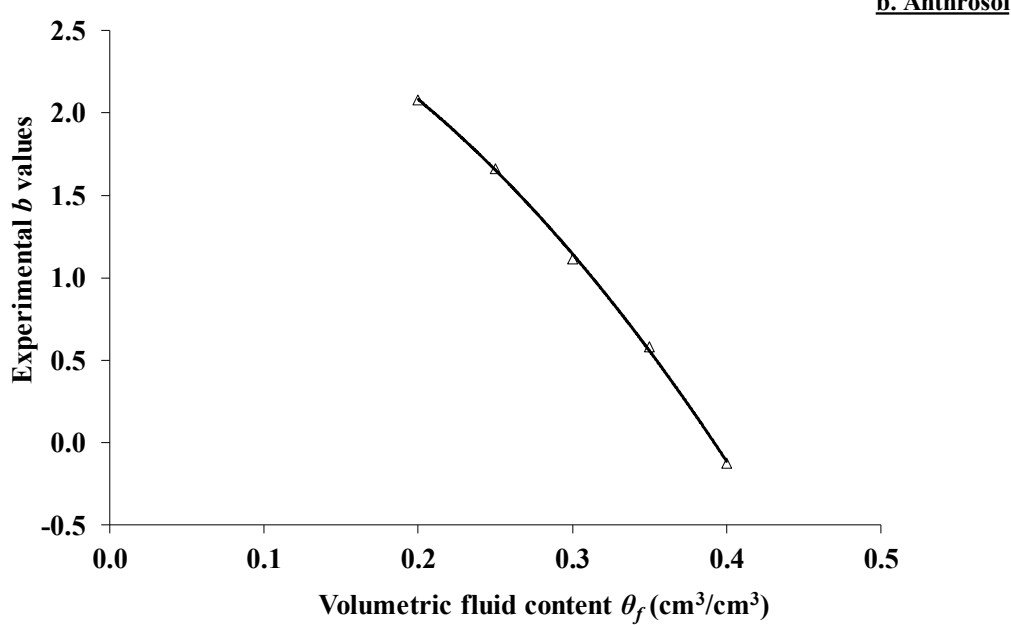

**Figure 4.** Experimental $b$ values of the $\beta$-ln($EC_b$) relationships versus volumetric fluid content ($\theta_f$): a) *Eutric Cambisol* and b) *Anthrosol*.

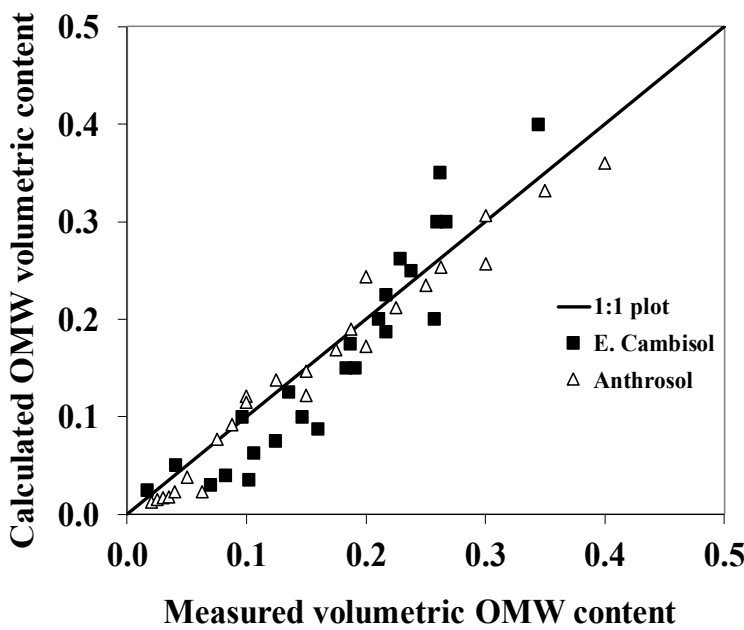

**Figure 5.** Calculated (equation 7) versus measured volumetric OMW content ($\theta_{OMW}$) for the two contaminated soils.