# Peer review of "Time domain reflectometry (TDR) for dielectric characterization"

_Hydrology and Earth System Sciences, 2020_

## Referee Comment (RC1) · Anonymous Referee #1 · 5 Mar 2020

The paper topic is interesting but the scientific approach is far to be robust, at the present stage. In fact, in the opinion of the reviewer, there are some major flaws and missing aspects that the Authors are strongly recommended to address before considering a possible resubmission. First of all, the models considered for the permittivity and for the (static) conductivity are very trivial and they do not take into account different non-ideal effects. Also, the literature overview does not include the works available at the state of the art, which propose accurate dielectric models and the related applicability. More importantly, the experiments conducted only on two soil samples are definitely poor and trivial. A comprehensive and more robust approach to the topic should be used, considering various materials, differing in qualitative characteris-

tics and also evaluating the corresponding metrological performance of the proposed method in terms of capability of discriminating the dielectric characteristics of different samples with respect to standard "reference" soil samples. As a consequence, the sensitivity and discrimination performance of the method should be analyzed and clearly defined.

―――――――――――――――――――

---

## Referee Comment (RC2) · Anonymous Referee #2 · 6 Mar 2020

The paper presents a methodology for a practical application of TDR to identify olive mill wastewater contaminated soils. I find the method to have potential, but, as most empirical models, it has some significant limitations. I encourage the authors to address and discuss these limitations in the revised manuscript.

1) Because this model is based only on EC, in field applications, it seems that other more benign sources of salinity may be confused for olive mill wastewater (OMW). What are the constituents in OMW that cause it to increase the TDR-measured bulk EC - only inorganic salts? On line 105, you mention that polyphenols were measured, but do not present these results in the paper - why? Are polyphenols relevant to this

model?

If it is only dissolved salts in the OMW, then the methodology you developed is best applied as a OMW contamination screening tool to identify soils that are potentially contaminated (i.e., contamination requires confirmation by other tests), rather than a direct OMW detection tool. In other words, more discussion on the limitations of this model is required.

2) I think it is possible that another approach to modeling your dataset was presented by Rhoades (see citations below) or potentially Hillhorst (2000). I encourage the authors to explore the similarities of their empirical model with other theoretical or semi-empirical models available in the literature. There is a possibility to quantify the increase in bulk soil EC to the EC of the OMW itself.

Rhoades, J. D., P. A. Raats, and R. J. Prather. 1976. Effects of liquid-phase electrical conductivity, water content, and surface conductivity on bulk soil electrical conductivity. Soil Sci. Soc. Am. J. 40:651-655.

Rhoades, J. D., N. A. Manteghi, P. J. Shouse, and W. J. Alves. 1989. Soil electrical conductivity and soil salinity: new formulation and calibration. Soil Sci. Soc. Am. J. 53:433-439.

Hilhorst M.A. 2000. A pore water conductivity sensor. Soil Sci. Soc. Am. J. 64:1922-1925.
* * *

---

## Author Comment (AC1) · 17 Apr 2020

Dear Referee #1,

With reference to the paper: hess 2020-69, by A. Comegna et al., please find below the replies to your review. Overall, the authors beg to disagree on a number of major points.

In particular, the Referee claims that the model adopted is "...very trivial...", a consideration which we found hard to fathom. We would like to point out that the model in question is the well-known exponential model based on Birchak's mixture equation

(Birchak et al., 1974), which was reformulated for our purposes. The model has been widely used, in the last 40 years, in soil physics (Dobson et al., 1985; Alharthi et al., 1986; Roth et al., 1990; Knight, R. and Endres, A., 1990; Heimovaara, T. J, 1994; Redman and DeRyck, 1994; Hilhorst, 1998; Knight, 2001; Chenaf and Amara, 2001; Persson and Berndtsson, 2002; Regalado et al., 2003; Huisman et al., 2003; Haridy et al., 2004; Mohamed and Said, 2005; Rinaldi and Francisca, 2006; Moroizumi and Sasaki, 2008; Francisca and Montoro, 2012; Zhan et al., 2013; Comegna et al., 2013a; Comegna et al., 2013b; Comegna et al., 2016, Comegna et al., 2017; Comegna et al., 2019; etc. . .).

Furthermore, the Referee states that: ". . .the literature overview does not include the works available at the state of the art". We agree that the literature in question is extensive. It was precisely for this reason that we included in the paper more than 30 references, which were selected from others for their affinity with our research.

Besides, according to the Referee the ". . .scientific approach is far to be robust. . .", which, objectively speaking, does not seem true to the facts. From an experimental point of view: i) two soils, pedologically different, were selected, ii) 160 measurements, for a full factorial analysis, were carried out only for model calibration and validation, iii) three different statistical indices (mean absolute percentage error: MAE, model efficiency: EF, and maximum absolute percentage error: ME) were calculated to assess model performance. Finally, the methodology developed in this study required statistical analysis of the relationship between the relative volume of OMW in water ($\beta$) and ECb, to determine the coefficients ac, b1, b2, b3 of equation (7). The analysis was performed using analysis of covariance (ANCOVA, at a significance level of 0.05), which is a statistical tool used to test the main and interaction effects of categorical variables on a continuous dependent variable, controlling for the effects of selected other continuous variables that co-vary with the dependent variable. Thus, the ANCOVA analysis combines regression analysis and analysis of variance, providing for each soil investigated a way of statistically controlling, in our case, the parallelism of the empirical

linear relationships $\beta$-ECb (figures 3a, b) observed at different levels of soil saturation $\theta$f. The results of the ANCOVA test, coupled with the other statistical indices computed are enough to validate, in our opinion, the methodology developed, whose results are indisputably encouraging (see figure 5).

Finally the Referee says: "More importantly, the experiments conducted only on two soil samples are definitely poor and trivial". It should be stressed that the soils selected belong to two typical pedological units of southern Italy which account for approximately 90% of the Italian olive plantations and the highest concentration of the olive oil industry in Italy. In these districts, despite European and Italian laws, there is the controversial propensity to spread olive mill wastewater (OMW) on soils, causing critical environmental problems.

The current database should of course be extended, selecting for example other pedological contexts in other Mediterranean countries sharing the same environmental problems. This aspect goes beyond the scope of the present research.

Sincerely

The authors

---

## Author Comment (AC2) · 17 Apr 2020

Dear Referee #2, With reference to the paper: hess 2020-69, by A. Comegna et al., please find below the replies to your review. The authors would like to thank the anonymous Referee for his useful suggestions which have been fully accepted. We explain below how the revised paper was reorganized.

Question 1

General premises

Solute monitoring in the soil can be achieved through the analysis of the variation of a

certain physico-chemical property that is able to characterize the solute concentration. Methods commonly used to characterize contaminated sites involve soil drilling, sampling, and the installation of monitoring wells for the collection of soil and water samples (Mercer and Cohen, 1990). Given the cost of these technologies, other noninvasive techniques that belong to geophysical methods have been sought to characterize contaminated sites extensively. Olive mill wastewater (OMW), like other contaminants (such as for e.g. hydrocarbons), is a very complex mixture made of different elements (Mahmoud et al., 2010; Piotrowska et al., 2011; Caputo et al., 2013, Mohawesh et al., 2014; Sahraoui et al., 2015; among others). Because of this composite nature, OMW detection can be a very difficult task. OMWs are known to be wastewaters rich in salts (especially potassium, calcium and magnesium). For this reason one may imagine to relate the concentration of a selected salt to the whole OMW amount in the soil. This approach is to be considered impractical for at least two reasons: i) soils may in turn contain a background concentration of the candidate salt. As a consequence, this background concentration must be determined in order to avoid mis-estimating the OMW final concentration, ii) the OMW composition changes in accordance with the quality of the olives, the type of maceration and the type of solvents used for cleaning the machines. Thus, again exactly as for the soil, chemical analysis of the wastewater should be carried out, from time to time, to establish the initial salt concentration. In order to overcome these difficulties, in this study we developed a general methodology, based on a dielectric approach for evaluating OMW presence in a contaminated soil. The methodology does not need, a priori, for the soil and OMW chemical composition to be known. That said, since OMWs are characterized by very high values of electrical conductivity (ECsol), we selected, on the basis of several laboratory tests, electrical conductivity as a candidate dielectric parameter for our approach, being furthermore easy to detect via the time domain reflectometry (TDR) technique. We explain in the following how the experimentation was fully implemented. A series of preliminary experiments (data not shown in the paper) were conducted in order to characterize the OMW dielectric response. Laboratory tests, in which simultaneous measurements

were made of dielectric permittivity $\varepsilon$sol and electrical conductivity ECsol, were carried out on solutions that were suitably prepared as a mix of known amounts of OMW and water. Measurements were carried out via the TDR technique; electrical conductivity was also measured using a Cyberscan conductivity meter (model 500). Data are shown in table 1 as a function of the relative volume of OMW in water $\beta$:

$\beta$=$\theta$OMW/(($\theta$w+$\theta$OMW))=$\theta$OMW/$\theta$f

where $\theta$f, $\theta$OMW and $\theta$w are respectively the volumetric content of the whole fluid phase, the volumetric OMW content and the volumetric water content. As deducible from table 1 (see below), it can be seen that the permittivity changes slightly with $\beta$, and that for $\beta$>0.30, $\varepsilon$sol cannot be estimated because the reflection point completely disappears (figure 1, see below). On the contrary, ECsol changes significantly, with a linear dependence, in the 0<$\beta$<1 domain (figure 2, see below). Starting from the above experimental evidence we further investigated in depth the OMW behavior in contaminated soil samples. From an experimental point of view, with reference to two distinct soils, 160 measurements, for a full factorial analysis, were carried out for model calibration and validation. Several statistical indices (MAE, EF, ME) were calculated, and finally a statistical ANCOVA test was performed at a significance level of 0.05 so as to consolidate from a statistical point of view the proposed methodology.

Specific comments

1a and 1c) In accordance with the Referee's comments contained in question #1, it should be emphasized that the soils selected belong to two typical pedological units of southern Italy which account for approximately 90% of the Italian olive plantations and the highest concentration of the olive oil industry in Italy. In these districts, despite European and Italian laws, there is the controversial propensity to spread olive mill wastewater (OMW) on soils, causing critical environmental problems. Under these considerations, in addition to what has already been stated in the general premise, one may assume that, in such field conditions, the main source of high electrical conductivity values is exclusively due to OMW.

1b) Line 105: polyphenols are known to be constituents of OMW. Their concentration can be relevant, and may cause, of course, several problems to soil biota. In the present paper, in order to characterize, from a physico-chemical point of view, the OMW used in the experimentation, we reported in table 2 (of the paper) the concentration of "Total polyphenols": "Total polyphenol content, in OMW, was obtained using the Folin-Ciocalteu colorimetric method".

Question 2

The soil extract method for salinity measurements, as proposed by US Salinity Laboratory Staff (1953) is laborious and destructive. New devices have been developed such as TDR (Topp et al., 1980), which exploit the variation in the dielectric behavior that a medium exhibits in the presence of an electromagnetic field. This behavior is described by the dielectric permittivity which is in general a complex function. The TDR technique is widely employed in Soil Physics to predict the volumetric water content ($\theta$) of a soil through the dielectric permittivity ($\varepsilon b$). TDR also allows us to estimate the soil bulk electrical conductivity (ECb) of the multiphase medium. The electrical conductivity of the bulk soil ECb is of course strongly related to the electrical conductivity of the soil solution (ECw) by means of empirical or theoretical correlations. Such correlations have been extensively investigated, notably by Rhoades et al. (1976), Rhoades et al. (1989), and Hilhorst (2000). Hilhorst (2000) presented a linear model relating $\varepsilon b$ and ECb in the form $\varepsilon b = AECb + B$, where $A = \varepsilon w / ECw = 80/ECw$, and $B = K0$ is the intercept of the line $\varepsilon\_b = f(ECb)$. Hilhorst concluded that his model could be validated for water contents up to saturation and for ECw values up to 0.3 S/m. He found that K0 depends on soil type and varies between 1.9 and 7.6. He recommended the value of 4.1 as generic offset. Many researchers (Hamed et al., 2003; Regalado et al., 2007; Persson 2002, among others) applied the deterministic model of Hilhorst to their experiments to convert ECb to ECW but did not use the same K0 offset value to achieve their study objectives. Nevertheless, K0=4.1 was proposed as a representative value for all soil

types. The Hilhorst's model must be applied for cases where $\theta>0.1$ since the conceptual model does not include the contribution due to ions moving through the lattice of ionic crystals in dry or almost dry soils. This same reason renders the above model inappropriate for the cases where bound water might be present, as may happen, for instance, in clays. Hilhorst's model appears to be a rather simplistic special case of the model reported by Rhoades et al. (1976), and Rhoades et al. (1989), in the sense that it does not take into account that $\varepsilon_b$ might be affected by the surface conductance of the soil matrix. The other most practical model of Rhoades et al. (1989) is based on the dual pathway parallel conductance (DPPC) approach and is applicable in open fields. The DPPC model demonstrated that $EC_b$ can be reduced to a nonlinear function of five soil properties: i) salinity as measured by the electrical conductivity of the saturated extracts ($EC_e$), ii) the saturation percentage (SP), iii) the volumetric soil water content ($\theta$), iv) the soil bulk density (b), and v) the soil temperature (t). These parameters must be kept in mind when interpreting $EC_b$ data. We conclude that the DPPC model is not easy to parametrize, and together with the Hilhorst model is subject to many limitations. We would like to say that referee's observations have been resolved in the revised version of the manuscript (abstract, section 3, and conclusions).

Sincerely

The authors

Please also note the supplement to this comment:
https://www.hydrol-earth-syst-sci-discuss.net/hess-2020-69/hess-2020-69-AC2-supplement.pdf
* * *
**Table1.**

| $\beta$ | $\varepsilon_{sol}$ | $EC_{sol}$ (TDR) (dS/m) | $EC_{sol}$(Cyberscan) (dS/m) |
|---|---|---|---|
| 0 (distilled water) | 78.89 | 0.013 | 0.012 |
| 0.10 | 77.86 | 1.45 | 1.52 |
| 0.20 | 75.77 | 2.76 | 2.79 |
| 0.30 | 73.46 | 3.90 | 3.86 |
| 0.40 | n.a. | 4.97 | 4.99 |
| 0.50 | n.a. | 5.95 | 6.02 |
| 0.60 | n.a. | 6.81 | 6.78 |
| 0.70 | n.a. | 7.75 | 7.70 |
| 0.80 | n.a. | 8.66 | 8.61 |
| 0.90 | n.a. | 9.25 | 9.20 |
| 1.0 (OMW) | n.a. | 10.20 | 10.18 |

[Figure]

**Figure 1.** Reflection coefficient versus time, for different $\beta$ values.

[Figure]

**Figure 2.** Electrical conductivity ($EC_{sol}$) versus $\beta$ values.